# Association between maternal mental health and early childhood development, nutrition, and common childhood illnesses in Khwisero subcounty, Kenya

**Amanuel Abajobir** \*, **Daniel Maina**, **Elizabeth Wambui**, **Estelle M. Sidze**

African Population and Health Research Center, Nairobi, Kenya

\* aabajobir@aphrc.org

## Abstract

### Background

Despite the significant public health burden of maternal mental health disorders in sub-Saharan Africa (SSA), limited data are available on their effects on early childhood development (ECD), nutritional status, and child health in the region.

### Aims

This study investigated the association between maternal mental health and ECD, nutritional status, and common childhood illnesses, while controlling for biological, social, financial, and health-related factors and/or confounders.

### Method

As part of the Innovative Partnership for Universal and Sustainable Healthcare (*i*-PUSH) program evaluation study, initiated in November 2019, a cohort of low-income rural families, including pregnant women or women of childbearing age with children under five, was recruited for this study. A total of 24 villages were randomly selected from a list of villages near two health facilities. Following a census to identify eligible households, 10 households per village were randomly selected. Data collection included maternal mental health, assessed using Centre for Epidemiological Studies Depression (CES-D) scale, ECD, nutritional status (anthropometric measurements), and common childhood illnesses, their symptoms, and healthcare utilization. This study presents a cross-sectional analysis of the data drawn from endline survey of 299 target mothers and 315 children.

### Results

The majority of the mothers were aged between 25 and 34 years. The mean age of children was 3.2 years, with 53% being male. The overall maternal mental health score, as measured by the CES-D scale, was 28. Children of mothers with higher CES-D scores exhibited poorer ECD domains, lower nutritional status indicators, and increased incidence of ill-

**Data Availability Statement:** All relevant data are within the paper and its Supporting information files.

**Funding:** The main study was funded by Dutch National Postcode Lottery, the Joep Lange Institute, and the Dutch Ministry of Foreign Affairs, and EDCTP2 programme supported by the European Union and Fondation Botnar (grant number TMA2020CDF-3101-i-PUSH-RCT); and supported by the. The views and opinions of authors expressed herein do not necessarily state or reflect those of EDCTP and Fondation Botnar. The funders had no role in study design, data collection and analysis, decision to publish, or preparation of the manuscript.

**Competing interests:** The authors have declared that no competing interests exist.

health in the previous two weeks, in both unadjusted and adjusted analyses. Individual, parental, and household factors—including maternal age, household wealth index, and decision-making regarding child healthcare—were significantly associated with children's development, nutrition status, and health outcomes.

## Conclusion

Children of mothers with low mental health scores demonstrated suboptimal developmental outcomes, nutritional status, and overall well-being, particularly for those from impoverished households. These findings suggest that improving the socioeconomic conditions of low-income households is essential for promoting children's development, nutritional status, and well-being. Longitudinal studies are needed to further investigate the impact of maternal mental health on child development, nutrition, and health outcomes, considering additional factors across the maternal, newborn, and child health continuum.

## Trial registration for the parent and nested study

ClinicalTrials.gov (NCT04068571), AEA Registry (AEARCTR-0006089) and PACTR (PACTR202204635504887).

## Introduction

The disease burden associated with maternal mental disorders is increasing, primarily among young and reproductive age women [1]. For instance, depression accounts for 41% of all disability-adjusted-life-years globally [2]. Poor maternal mental well-being has consistently been implicated as a cause of adverse developmental outcomes, poor nutritional status, and suboptimal health in children, often resulting in long-lasting consequences [3–6].

In low- and middle-income countries (LMICs), including the SSA, the burden of maternal mental disorders (e.g., perinatal depression), ranges from 19% to 28%, with their impact on child development and health emerging as a high-priority concern [7, 8]. For instance, mothers experiencing depression during this critical period are more likely to have offspring with poor developmental outcomes [9, 10], suboptimal physical growth, stunting, and underweight [8–15], as well as smaller head circumferences at birth [10]. Additionally, studies indicate that factors such as inadequate maternal self-care, poor feeding practices—including not exclusively breastfeeding or early cessation of breastfeeding—suboptimal parenting, inadequate nursing, poor hygiene, and inadequate healthcare access are associated with detrimental child developmental and health outcomes [12].

There is a significant lack of data on maternal mental health and limited evidence regarding feasible screening and treatment strategies in LMICs. There is also lack of specialized services and guidelines tailored for expectant and nursing mothers. Stigmatizing attitudes from primary healthcare providers and the public are also common [6]. In Nigeria, for instance, maternal healthcare facilities are characterized by inadequate resources to provide mental health services, resulting in a significant treatment gap for women experiencing perinatal depression [16]. Up to 33% of adolescent mothers in Kenya experience perinatal depression [14, 15], which adversely affects their overall health [17]. This is primarily due to limited access to basic healthcare, stemming from barriers pertaining to infrastructure as well as social and cultural norms. Such adversities are likely to have negative consequences on maternal mental health,

including prolonged periods of insecurity and depression following childbirth [14, 15]. These mental health challenges affect childcare practices [9] and subsequently influence the developmental outcomes for their children.

Sociodemographic and structural factors associated with poor maternal mental health include teenage motherhood, poverty and economic disadvantage, unemployment, marital status (divorce, death of a spouse or lack of a supportive partner), stressful life events, intimate partner violence, ill-health, lifestyle factors, lack of social support, and food insecurity [4–15]. Evidence of reverse causation has been reported, indicating that mothers of malnourished children exhibit higher rates of moderate to severe depression compared to mothers of children with normal weight [5]. However, these associations are inconsistent. For instance, certain studies have found no significant differences in growth indicators between children whose mothers experience perinatal depression and those whose mothers do not [18].

Little is also known on the effect of maternal mental health on child development, nutrition and health in LMICs, despite the presence of numerous risk factors at the individual, household, and environment levels, as well as factors specifically related to mothers and children [7, 16]. The effects of maternal mental disorders on child developmental outcomes have garnered limited attention in SSA, and the existing literature is inconsistent. There is scarcity of a prospective community-based cohort studies capable of quantifying modifiable conditions, despite evidence indicating that the rates of poor development, nutritional deficiencies, and health issues in offspring of mothers with mental health disorders are significantly higher than those of children of mothers without such disorders [7–9, 13, 18]. Consequently, further research is essential to elucidate the relationship between maternal mental health and ECD, nutritional status, and health outcomes in low-income mothers and/or households in semi-rural settings, while controlling for potential confounders.

## Methods

### Study context

With the exception of the ECD data, most of the data used in this analysis were obtained from the endline survey conducted as part of the evaluation of the Innovative Partnership for Universal and Sustainable Healthcare (*i*-PUSH) program. The survey focused on women from socioeconomically disadvantaged households in Khwisero Sub-county, Kenya ("parent" study hereafter) [19]. The "parent" study's protocol was registered (AEA Registry [AEARCTR-0006089] and ClinicalTrials.gov [NCT04068571]). The nested study was also registered (PACTR202204635504887).

### Study design

The original survey design for the "parent" study was a longitudinal cluster randomized controlled trial (RCT), with randomization occurring at the village level (more details are found elsewhere [19]). However, this study used data exclusively from a single data collection point (endline) due to the limited sample size for the outcome variables. The recruitment process for the "parent" study started on 16/11/2019, and ended on 11/11/2021, specifically for the collection of ECD data. Consequently, this study entailed a cross-sectional analysis of data collected during the endline survey, focusing on target mothers and their children.

### Recruitment of study participants

This research employed data derived from the *i*-PUSH program evaluation study, which commenced in November 2019. The study began by enrolling a cohort of rural income households,

where participants included pregnant women or women of reproductive age with children under 4 years old. A total of 24 villages were randomly selected from lists of 239 villages surrounding two different health facilities. After conducting a census in each selected village to identify all the households meeting the aforementioned requirements, 10 households were randomly selected from each village. Within each household, adults, including the target women, reported on ECD and various health-related events concerning the children, including symptoms and healthcare service utilization. Additionally, anthropometric measurements were performed to assess nutritional status of the children. For this analysis, we specifically used data obtained during the endline assessment for program evaluation, as baseline data collection only captured a limited number of child health events and measurements.

## Study population and sample size

As per the "parent" study, the study population consisted of all potentially eligible households residing in the selected villages, specifically targeting those with at least one woman of reproductive age (WRA), aged 18 to 49 years, who was either pregnant or had at least one child under four-year-old at baseline. The sample size computation fixed the number of clusters per arm at 12 clusters, which facilitated the determination of the cluster size and the overall sample size [18] for the evaluation of the *i*-PUSH program's impact, projected to produce an impact size of 0.4 regarding healthcare utilization, supported by an intracluster correlation (ICC) of 0.014. The estimates of the ICC were taken from a study conducted in Nandi County (Kenya) that used high-frequency data on financial expenditures and healthcare-seeking behaviors over a year [20]. Regardless of reported health symptoms, visits to any formal healthcare facility were used to construct the ICC as a proxy for healthcare utilization. The calculation used an 80% power, a 5% error margin, and a 95% confidence interval (CI) [21]. There were 12 clusters and 10 women in each cluster per arm. With 12 evenly sized clusters in each arm, 10 participants in each cluster, and a cluster size of 121/12 = 10, a total sample size of 120 women was calculated for each arm, with a total sample of 240 women. The detailed sample size calculation for the "parent" study is found elsewhere [18]. For this analysis, data were collected from a total of 299 women and 315 children who participated in the endline survey. It is worth noting that in some households, there were multiple WRA, which resulted in a higher number of women than eligible households. Additionally, a small number of participating women served as caretakers for the children but were not biologically related to them. The role of these non-biological caretakers alternated between biological mothers, making it difficult to analyze separately given the overlapping caregiving roles and the low number of such cases.

## Outcome variables

**ECD.** The ECD data collection tool was adopted using validated tools in LMICs, including Kenya's rural and urban cultural settings [20–22]. Mothers with children aged 8–60 months old (at the endline survey) were interviewed using this tool, which measured various developmental milestones in cognitive development (6 items), language and general communication development (79 items), self-help/adaptation (11 items), socio-emotional and personal-social development (11 items), and emotional development/self-regulation (25 items). The sum of the scores for each item in each domain was used to determine the final score in each developmental domain. Children that scored above the mean were likely to have optimal developmental outcomes [20, 21]. The scores for these variables ranged from 0–18, 1–237, 0–33, 7–33, and 14–75, respectively, for cognitive development, language and general communication, self-help/adaptation, socio-emotional and personal-social development, and emotional development/self-regulation. Table 1 provides a summary of this outcome variable, which comprised

composite scores produced based on the categories mentioned above. For more details, see the S1–S3 Tables in S3 File.

**Nutrition.** Anthropometric measurements, such as length/height, weight, and mid-upper arm circumference (MUAC) measurements were obtained by qualified quantitative field interviewers who were trained in anthropometric data collection techniques with the assistance of community health volunteers (CHVs) in order to determine the nutritional status of children under the age of five at the endline survey. The World Health Organization's (WHO's) Child Growth Standards were then used to construct anthropometric indicators [23, 24]. These comprised length/height-for-age z-score (stunting), weight-for-age z-score (underweight), weight-for-length/height z-score (wasting) and MUAC-for-age z-scores. The measurements obtained exhibited a range of 59 to 124cm for length/height, 1,300 to 24,450gm for the weight of the child, and 10.00 to 19.50cm for MUAC. To determine how much and in which direction each child's measurement deviated from the mean of the WHO's Child Growth Standards, z-scores were computed [23]. Outside of the normal range (-2 to +2) z-score values signify a specific nutritional problem, such as stunting, wasting, underweight, or overweight. A summary of these measurements and the indicators based on WHO's Child Growth Standards is given in Table 1.

**Health.** Mothers and/or primary caregivers were asked to report any illnesses experienced by children under 5 years of age within the last two weeks prior to the baseline, midline, and endline surveys. Respondents were also asked to report the specific illnesses their child(ren) had, which included malaria, pneumonia, cold/flu, diarrheal disease, and any other illnesses (e.g., eczema/contact dermatitis, etc.). All of these outcomes were responded on a binary (Yes/No) scale. Outcome variables such as diarrheal disease, pneumonia and eczema/contact dermatitis were excluded from the analysis as their numbers remained scanty.

## Predictor variable

The study used a modified self-reporting 20-item CES-D scale [25, 26] to collect data on mother's (or female caregiver's) mental health at the endline survey (June 2021). Ten items on four scales were used to ask participants to rate how often they had conflicting feelings over the course of the past seven days on four scales: (1) never; (2) a little of the time (1–2 days); (3) a moderate amount of the time (3–4 days); and (4) most or all of the time (5–7 days). In addition, participants were asked to rate their level of agreement with a statement using 10 items on five scales: strongly agree, agree, agree, undecided, disagree, and strongly disagree. Statements such, "I can usually achieve what I want if I work hard for it"; "once I make plans, I am almost certain to make them work"; etc. were used. The final score was calculated by adding the 20 items together. Although any score equal to or over 16 denotes "poor mental health" or depression on the CES-D scale, and is reliable and valid in diverse African settings, we utilized the mean score since mental health disorders often happen in a continuum rather than in discrete episodes [25–27]. The scores for this variable ranged from 8 to 45. The analysis employed mental health data collected at the endline since the analysis's power was reduced by the small number of cases for outcome variables at the baseline or midline data collection waves.

## Confounders

Given that maternal mental health disorders and their effect on ECD, child nutrition, and health are influenced by various contextual factors, such as family/household socioeconomic status, wealth, availability of formal and informal supports, health literacy, etc., our analyses accounted for these variables. Child covariates included sex (male/female) and age (in years). Parental variables comprised mother's age (in years), literacy, education, income, marital

**Table 1. Description of study variables.**

| Variable | Frequency* (%) |
|---|---|
| Sex of child | |
| Male | 159 (53.2) |
| Female | 140 (46.8) |
| Age of child (years) | 3.20 [0.32, 5]** |
| Maternal age (years) | |
| 15–24 | 88 (30.0) |
| 25–34 | 137 (46.6) |
| 35–44 | 69 (23.4) |
| Maternal educational status | |
| Primary not completed | 188 (66.9) |
| At least primary | 93 (33.1) |
| Maternal literacy | |
| Cannot read and write | 13 (4.5) |
| Can read and write | 274 (95.5) |
| Mother's income | |
| Not own income | 132 (46.2) |
| Own income | 154 (53.8) |
| Maternal marital status | |
| Never married | 19 (6.7) |
| Married | 236 (83.1) |
| Cohabiting | 19 (6.7) |
| Divorced/separated/widowed | 10 (3.5) |
| Maternal history of any chronic disease | |
| No | 111 (71.2) |
| Yes | 45 (28.8) |
| Presence of father in the household | |
| Father lived together/household member | 8 (2.8) |
| Father was not household member but lived in the same community | 11 (3.8) |
| Father was not a household member and not living in the same community | 162 (56.6) |
| Father was dead | 105 (36.8) |
| Household wealth terciles | |
| Poor | 17 (5.8) |
| Middle | 131 (44.6) |
| Rich | 146 (49.6) |
| Child health-seeking decision | |
| Mother alone | 120 (44.3) |
| Husband/partner/head alone | 50 (18.5) |
| Husband/partner/head together | 98 (36.2) |
| Another household member | 3 (1.0) |
| Paid care | |
| No | 245 (89.4) |
| Yes | 29 (10.6) |
| Maternal CES-D score | 28 [8, 45]** |
| Cognitive development | 11 [0, 18]** |
| Language and general communication | 137 [1, 237]** |
| Self-help/adaptation | 21 [0, 33]** |
| Socio-emotional and personal-social development | 21.4 [7, 33]** |

(*Continued*)

Table 1. (Continued)

| Variable | Frequency* (%) |
|---|---|
| Emotional development/self-regulation | 36 [14, 75]** |
| Length/height (cm) | 92 [59, 124]** |
| Weight of the child (grams) | 13,582 [1,300, 24,450]** |
| MUAC (cm) | 15.72 [10.00, 19.50]** |
| Child was unwell in the last 2 weeks | 153 (48.6) |
| Malaria | 61 (19.4) |
| Cold/flu | 101 (32.0) |

*Since missing observations were excluded from the analyses, the number of observations for each variable may vary.
**Mean, cumulative score and range.

status, the presence of the father in the household, and the household wealth index. Additionally, the analyses accounted for maternal history of chronic illnesses, decision making regarding children's health care at the household level, and the availability of paid childcare facilities/services. The wealth index of households was calculated using data on asset ownership within the household.

## Data management and analysis

There were strict data quality control measures in place, such as: pre-test to fix any issues that arose with the tools, the participants, or the field environment. Real-time data collection in the field was supervised by trained team leaders, who also ensured data quality by conducting frequent spot checks and sit-ins on up to 5% of each field worker's daily work. By editing the data before they were recorded, they also validated the accuracy of the data collected before sending it to the database. Additionally, an automated tool verified the accuracy, consistency, and completeness of the data.

To identify any similarities and/or variations in participant characteristics between the groups, the first set of analysis comprised descriptive statistics, which summarized and contrasted the data using measures of central tendency and dispersion. The descriptive statistics compared some baseline or endline basic characteristics using chi-square for binary variables and a t-test for continuous variables. A series of logistic regression analyses were conducted in the second set of the analyses to ascertain the association between predictor and outcome variables. This process involved fitting an unadjusted (univariate) model that considered each outcome and predicator one at a time, followed by an adjusted (multivariate) model that incorporated all covariates at once. Distinct regression models were fitted for each outcome. The analyses were done using R statistical package and the results were presented as prevalence odds ratios (POR) along with 95%CI.

## Ethical considerations

Participants were given a written informed consent form describing the objectives of the study and requesting their consent to sign and participate in the study. The information sheet sent to participants included information on the purpose of the study, key project details, potential risks and benefits for participants, expectations for privacy and confidentiality, and contact information. Each participant was free to choose whether or not to participate in the study, and they were also given the option to opt out. Young women under the age of 18 (n = 9) who were pregnant or nursing a child were given consent forms by trained fieldworkers since they were deemed emancipated minors (able to give their own consent). If they were unable to

write, they were given a stamp and/or a witness to sign and confirm the consent, and this was approved by the ethics review board. Each participant received an anonymous number or ID, and the information we obtained from them was kept private. The "parent" study's study protocol was reviewed and approved by Amref Health Africa's Ethical and Scientific Review Committee (ESRC) (ref. P679-2019). The nested study that served as the foundation for this analysis was also reviewed and approved by the ESRC (ref. P1060/2021). A research permit was granted by the Kenya's National Commission for Science, Technology, and Innovation (NACOSTI). The authors confirm that all procedures contributing to this work comply with the ethical standards of the relevant national and institutional committees on human participants and with the Helsinki Declaration of 1975 (2008 version). See details in the S1 and S2 Files.

## Results

The predominant age of the mothers ranged from 25 to 34 years. The mean age of the children was 3.2 years, and about 53% were males. Study participants remained similar across all basic characteristics. The mean scores of cognitive development and language and general communication, self-help/adaptation, socio-emotional and personal-social development and emotional development/self-regulation were 16, 137, 21, 21.4, and 36, respectively. The mean measures of length/height, weight and MUAC were 92cm, 13,582gm, and 15.72cm, respectively. The mean CES-D score for women in overall sample was 28 (Table 1).

### Maternal mental health and ECD

The relationship between maternal mental health and ECD, as well as child nutrition and health outcomes, is presented in Table 2. Although a decrease in the effect size of ECD across all domains was observed for children whose mothers exhibited higher CES-D scores, no statistically significant differences in ECD scores on the unadjusted and adjusted analyses. Additionally, some effect size values indicated a negative correlation between high CES-D scores and the majority of ECD domains.

**Table 2. The correlation between maternal mental health score, and ECD domains, nutritional indicators, and well-being.**

| ECD domains | Unadjusted | | | Adjusted* | | |
|---|---|---|---|---|---|---|
| | POR | 95% CI | p-value | POR | 95% CI | p-value |
| Cognitive development | 1.06 | 0.93, 1.21 | 0.40 | 1.09 | 0.84, 1.03 | 0.20 |
| Language development and general communication | 1.15 | 0.58, 2.27 | 0.70 | 1.04 | 0.51, 1.73 | 0.80 |
| Self-help/adaptation | 1.08 | 0.94, 1.23 | 0.30 | 1.02 | 0.91, 1.13 | 0.80 |
| Socio-emotional and personal-social development | 1.14 | 0.87, 1.02 | 0.15 | 1.13 | 0.84, 1.48 | 0.11 |
| Emotional development/self-regulation | 1.28 | 0.77, 1.91 | 0.07 | 1.09 | 0.77, 1.01 | 0.07 |
| **Nutrition outcomes/indicators** | | | | | | |
| Length/height-for-age z-score | 2.64 | 2.56, 2.70 | 0.003 | 2.61 | 2.53, 3.66 | 0.001 |
| Weight-for-age z-score | 2.66 | 2.54, 3.728 | <0.001 | 2.71 | 2.61, 4.76 | <0.001 |
| Weight-for-length/height z-score | 2.73 | 2.64, 3.81 | 0.031 | 2.71 | 2.63, 5.49 | 0.023 |
| MUAC-for-age z-score | 2.73 | 2.39, 3.77 | 0.002 | 2.72 | 2.36, 4.93 | <0.001 |
| **Child health outcomes** | | | | | | |
| Child unwell | 2.87 | 2.74, 3.00 | 0.009 | 2.98 | 2.08, 3.32 | 0.002 |
| Malaria | 2.80 | 2.64, 3.01 | 0.40 | 2.74 | 2.53, 3.00 | 0.80 |
| Cold/flu | 2.59 | 2.42, 2.83 | 0.10 | 2.61 | 2.39, 2.92 | 0.30 |

Moreover, the adjusted models failed to detect a meaningful relationship between maternal mental health scores and ECD outcomes. Interestingly, older children showed higher scores in their ECD outcomes across all domains both in unadjusted and adjusted models (e.g., cognitive development (adjusted POR = 2.94, p < 0.001); language development and general communication (adjusted POR = 1.69, p < 0.001); self-help/adaptation (adjusted POR = 3.39, p < 0.001); emotional development/self-regulation (adjusted POR = 2.51, p < 0.001). Although maternal age of 35–44 years, owning income by mothers, father's presence in the same household, child healthcare-seeking decisions by household heads, and having access to paid care significantly increased ECD scores in most of the domains in unadjusted models, although the significance differences waned after adjustment for other confounders (S1 Table in S3 File).

## Maternal mental health and child nutritional outcomes

Children whose mothers reported higher mental health (CES-D) scores exhibited decreased odds for all nutritional indicators in both the unadjusted and adjusted analyses. The adjusted models revealed a significant association between maternal CES-D scores and nutritional outcomes, with a negative association observed across all adjusted values. Older children displayed higher nutritional outcomes in all scores (e.g., weight-for-age z-score (adjusted POR = 0.86, p = 0.02) (S2 Table in S3 File).

## Maternal mental health and child well-being

Children of mothers with high CES-D scores had the odds of 1.05 and 1.09 of being unwell for the previous two weeks, respectively. The associations between maternal CES-D scores and child illnesses from malaria and the cold/flu were statistically insignificant (Table 2).

In both unadjusted and adjusted models, the overall well-being of children from households with higher wealth index was significantly higher. Malaria risk increased for children when decisions about their care were made solely by the spouse, head of the household, or partner, although it reduced for mothers between the ages of 34 and 45 (S3 Table in S3 File). All relevant data are within the manuscript and its Supporting Information files (see the raw datasets provided in Excel format documents).

## Discussion

The study found no statistically significant association maternal mental health (high CES-D scores) and ECD outcomes after adjustments, although a negative trend was observed across most ECD domains. Maternal CES-D scores were negatively associated with child nutritional outcomes, and child well-being significantly improved with higher household wealth, while malaria risk increased when care decisions were made solely by the household head or spouse. This study posited that children born to mothers with mental problems may exhibit adverse outcomes in childhood development, nutrition, and health. Although the findings indicated reduced ECD scores in children whose mothers had poor mental health, certain associations did not achieve statistical significance. A South African study suggested similarities in linguistic and cognitive development irrespective of maternal perinatal depression, emphasizing the need for extended longitudinal observations [9]. Similarly, delays in motor growth and speech parameters were reported, independent of perinatal depression [17]. Contrasting findings from another study in South Africa showed no association between maternal antenatal psychological distress and early developmental outcomes [28]. This study aligns with existing evidence, revealing factors such as maternal age, income, access to paid care, and the presence of

biological fathers as contributors to improved ECD outcomes and protective measures against poor maternal mental health [29].

The role of parental risks, including poverty, acquired both before and during pregnancy, is highlighted, influencing maternal mental disorders and childhood developmental outcomes through various mechanisms [30, 31]. Sociodemographic and structural determinants such as teenage motherhood, economic disadvantage, food insecurity, and lack of social support are identified in LMICs [7, 8, 18]. Consistent with prior evidence, this study emphasizes the significance of characteristics such as maternal age, income, access to paid care, and the presence of biological fathers in promoting positive ECD outcomes and mitigating poor maternal mental health [29].

Maternal mental disorders during the postpartum period are shown to impact maternal-infant bonding, leading to potential issues, including maternal over intrusiveness, emotional withdrawal, and lack of responsive caregiving [32]. The study underscores the critical influence of perinatal maternal mental health on child growth and development in the first one thousand days, emphasizing the need for early interventions. Existing strategies are critiqued for potentially commencing too late in development, emphasizing the crucial role of parental health, particularly maternal mental health, in laying a healthy foundation for the development of children.

This study also establishes an association between maternal mental health and the nutritional status of children under five. After adjusting for various factors, children of mothers with poor mental health demonstrated a higher likelihood of poor nutritional status [9]. Consistent evidence from SSA links maternal mental disorders to adverse developmental outcomes in young children, including nutritional outcomes and physical dimensions of growth. Notably, maternal antenatal psychological distress has been associated with smaller head circumference at birth, and lasting depression compromises physical growth and increases the risk of behavior problems [9, 32]. The study acknowledges inconsistencies in these associations, potentially influenced by unassessed structural and sociodemographic factors.

Additionally, this research explores the wider influence of maternal mental health on the overall well-being of children and their susceptibility to other infectious diseases associated with poverty, such as malaria and cold/flu, despite the absence of statistically significant associations. Studies on the intersection of maternal mental health and childhood illnesses are relatively scarce in SSA. The study contributes insights into the social, financial, and health factors influencing these associations, highlighting the role of maternal age, decision-making in child healthcare, and household assets in child well-being and illness incidence.

This study contributes to the growing body of evidence by examining the effect of maternal mental health on ECD and children's nutritional status, as well as overall well-being and ill-health from common childhood conditions. Based on the findings of this study, interventions targeting maternal mental health should be integrated into existing maternal and child health programs. Early identification and support for mothers experiencing mental health issues, particularly during the perinatal period, can contribute to improved childhood development, nutrition, and overall health outcomes. Programs should focus on addressing socioeconomic factors such as poverty, food insecurity, and lack of social support, which are identified as key determinants of maternal mental health and childhood outcomes.

However, the unexpected findings regarding the weak association between higher maternal CES-D scores and ECD can be attributed to several potential explanations. First, maternal mental health was assessed using a self-reported CES-D scale, which may introduce response bias; for instance, respondents might underreport or overreport symptoms [33], distorting results and weakening the possible correlations between maternal mental health scores and ECD outcomes. We also acknowledge the use of data collected at a single point in time

(endline) might excluded changes that could have occurred between baseline and the endline surveys, limiting observations on changes that may have happened over time. Additionally, despite adjusting for various covariates, residual confounders, including social support networks and other unmeasured environmental stressors, may have influenced the observed associations. Moreover, the weak association may be reflective of cultural variations in attitudes toward mental health and the availability/accessibility of mental health resources, which may modify the effect of maternal mental health on child development [33]. Lastly, variability in child outcomes may also be influenced by genetic attributes inherent to the children that are independent of maternal mental health status.

In conclusion, this study highlights the intricate associations between maternal mental health and various dimensions of childhood development, nutrition, and well-being. While some associations lacked statistical significance, the overall body of evidence underscores the importance of considering maternal mental health as a crucial factor in shaping the well-being of children. The study advocates for a holistic approach that encompasses not only the individual factors related to maternal mental health but also broader social, economic, and health determinants. In alignment with Sustainable Development Goal 3, the study emphasizes the need for sustained efforts to develop evidence-based strategies aimed at improving maternal and child health outcomes, especially in low- and middle-income countries. The findings call for longitudinal studies to further explore the modifying factors during the perinatal period, laying the groundwork for future strategies that enhance the health and well-being of mothers and children.

## Supporting information

**S1 File. P1060-2021 approval letter for renewal and amendments.**
(PDF)

**S2 File. P679-2019 approval letter.**
(PDF)

**S3 File. S1-S3 Tables.**
(DOCX)

**S1 Dataset. Dataset anthropometric outcomes.**
(XLSX)

**S2 Dataset. Dataset ECD outcomes.**
(XLSX)

**S3 Dataset. Dataset health outcomes.**
(XLSX)

**S4 Dataset. Dataset mental health.**
(XLSX)

## Acknowledgments

The research team is grateful to the funder, implementers, study participants, and field team, as well as county and sub-county officials and other stakeholders who contributed directly or indirectly to the study's success.

## Author Contributions

**Conceptualization:** Amanuel Abajobir.

**Data curation:** Daniel Maina.

**Formal analysis:** Amanuel Abajobir.

**Funding acquisition:** Amanuel Abajobir.

**Investigation:** Amanuel Abajobir.

**Methodology:** Amanuel Abajobir, Estelle M. Sidze.

**Project administration:** Amanuel Abajobir.

**Resources:** Amanuel Abajobir, Estelle M. Sidze.

**Software:** Daniel Maina.

**Supervision:** Amanuel Abajobir, Daniel Maina, Estelle M. Sidze.

**Validation:** Amanuel Abajobir.

**Visualization:** Amanuel Abajobir.

**Writing – original draft:** Amanuel Abajobir.

**Writing – review & editing:** Daniel Maina, Elizabeth Wambui, Estelle M. Sidze.

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
