## [Decision Letter · Decision Letter 0]

21 Aug 2024

PONE-D-23-42156Maternal mental health and early childhood development, nutritional status and common childhood illnesses in Khwisero subcounty, KenyaPLOS ONE

Dear Dr. Abajobir,

Thank you for submitting your manuscript to PLOS ONE. After careful consideration, we feel that it has merit but does not fully meet PLOS ONE’s publication criteria as it currently stands. Therefore, we invite you to submit a revised version of the manuscript that addresses the points raised during the review process.

We look forward to receiving your revised manuscript.

Kind regards,

Ammal Mokhtar Metwally, Ph.D (MD)

Academic Editor

PLOS ONE

“The study is funded by i-PUSH-RCT which is part of the EDCTP2 programme supported by the European Union (grant number TMA2020CDF-3101-i-PUSH-RCT); and supported by the Fondation Botnar. The views and opinions of authors expressed herein do not necessarily state or reflect those of EDCTP and Fondation Botnar.”

3. In the online submission form, you indicated that [Since they pertain to a "high-risk" population, the data are not generally accessible but will be provided upon request. Links to published full-text protocol, protocol registrations, and other resources are offered, nevertheless. The published protocol is available at: https://www.ncbi.nlm.nih.gov/pmc/articles/PMC8443110/.].

Additional Editor Comments:

The manuscript is interested meanwhile, the reviewers have raised a number of points which we believe would improve the manuscript and may allow a revised version to be published in PLOS one.

Reviewers' comments:

Reviewer's Responses to Questions

**Comments to the Author**

1. Is the manuscript technically sound, and do the data support the conclusions?

Reviewer #1: Partly

Reviewer #2: Yes

2. Has the statistical analysis been performed appropriately and rigorously? 

Reviewer #1: No

Reviewer #2: Yes

3. Have the authors made all data underlying the findings in their manuscript fully available?

Reviewer #1: No

Reviewer #2: No

4. Is the manuscript presented in an intelligible fashion and written in standard English?

Reviewer #1: Yes

Reviewer #2: Yes

5. Review Comments to the Author

Reviewer #1: Thank you for submitting this important work. Please consider the following recommendations:

1. Please indicate the research question for the parent study so readers can evaluate the relationship between the studies.

2. If that question is related at all to the research question of this study, then you should explore participation in the intervention group as a potential confounder.

3. Demographic issues: participation by women under 18 is unusual. Indicate how many minors participated. Also, how many women serving as caretakers but not related to the child participated in this study? Could that be confounding the association?

4. Were the ECD scores assessed by age group? Otherwise, it is normal for older children to show more advanced development than younger children. It doesn't make sense to analyze such a large age range (8-60 months) together using the same ECD measures for all of them. What you want to know is, is each child developmentally appropriate for their age. I don't see this info in the supplemental information.

5. Please include tables in the supplementary materials for all results reported for which you note, "Table not shown."

6. Methods: Typically, the effect measure for cross-sectional studies is a POR. It would be better / more useful to report and interpret the effect measures rather than simply report the beta coefficients.

7. Results section: In the third line, you say "for both groups." But this study has just one group.

8. Please report the possible ranges for all scores reported (perhaps when you describe the measures in the Methods section) because the reader cannot evaluate whether reported means are high or low, or otherwise clinically important.

9. Please rewrite the first paragraph of the Discussion to restate the main findings of the study and its significance. Then move into comparisons to other studies.

10. I'd like to see more discussion about why there were unexpected findings.

11. Please check the manuscript for correct English grammar, e.g., see the first sentence under "Confounders."

Reviewer #2: Overall a sound and thorough research paper with robust statistical conclusions.

Additionally, the history of childhood vaccinations recommended for first 2 years of life and maternal tetanus vaccinations that are given during or before pregnancy could have also been included as covariates.

Moreover, since this is a cross sectional study, instead of reporting B coefficients of logistic regression, Prevalence Odds ratios (POR) along with their confidence intervals and p values should be reported.

6. PLOS authors have the option to publish the peer review history of their article (what does this mean?). If published, this will include your full peer review and any attached files.

Reviewer #1: No

Reviewer #2: **Yes: **Dr. Ammar Ali Muhammad

---

## [Author Response · Author response to Decision Letter 0]

23 Oct 2024

Dear Dr Ammal Mokhtar Metwally Date: 18/10/2024

Academic Editor

PLOS ONE

Subject: Point-by-point Response Letter for Submission ID [PONE-D-23-42156] - [EMID:9d1817402dbca406]

We hope this letter finds you well. We appreciate the time and effort you and the reviewers have dedicated to assessing the second round of the above-referenced manuscript. We are grateful for the insightful comments and suggestions provided.

We have carefully considered each comment/suggestion, and we would like to assure you that all comments have been addressed meticulously and accordingly, one-by-one. Below, we provide a detailed response to the key points raised during the review process and how have addressed them.

We believe that the revisions made in response to the comments/suggestions have significantly strengthened the quality and clarity of the manuscript and supplementary materials. We are therefore confident that the changes address the concerns raised and enhance the overall contribution of our work to the field.

The main study was funded by Dutch National Postcode Lottery, the Joep Lange Institute, and the Dutch Ministry of Foreign Affairs, and EDCTP2 programme supported by the European Union and Fondation Botnar (grant number TMA2020CDF-3101-i-PUSH-RCT); and supported by the. The views and opinions of authors expressed herein do not necessarily state or reflect those of EDCTP and Fondation Botnar. The funders had no role in study design, data collection and analysis, decision to publish, or preparation of the manuscript.

Thank you once again for the constructive feedback, which has undeniably improved the manuscript. We look forward for your further consideration of this version of the manuscript in your esteemed journal. 

Sincerely,

Amanuel Abajobir (PhD), on behalf of all authors

---

## [Decision Letter · Decision Letter 1]

6 Jan 2025

Association between maternal mental health and early childhood development, nutrition, and common childhood illnesses in Khwisero subcounty, Kenya

PONE-D-23-42156R1

Dear Dr. Abajobir,

We’re pleased to inform you that your manuscript has been judged scientifically suitable for publication and will be formally accepted for publication once it meets all outstanding technical requirements.

Kind regards,

Ammal Mokhtar Metwally, Ph.D (MD)

Academic Editor

PLOS ONE

Additional Editor Comments (optional):

Reviewers' comments:

Reviewer's Responses to Questions

**Comments to the Author**

1. If the authors have adequately addressed your comments raised in a previous round of review and you feel that this manuscript is now acceptable for publication, you may indicate that here to bypass the “Comments to the Author” section, enter your conflict of interest statement in the “Confidential to Editor” section, and submit your "Accept" recommendation.

Reviewer #2: All comments have been addressed

2. Is the manuscript technically sound, and do the data support the conclusions?

Reviewer #2: Yes

3. Has the statistical analysis been performed appropriately and rigorously? 

Reviewer #2: Yes

4. Have the authors made all data underlying the findings in their manuscript fully available?

Reviewer #2: No

5. Is the manuscript presented in an intelligible fashion and written in standard English?

Reviewer #2: Yes

6. Review Comments to the Author

Reviewer #2: (No Response)

7. PLOS authors have the option to publish the peer review history of their article (what does this mean?). If published, this will include your full peer review and any attached files.

Reviewer #2: No

---

## [Editor Report · Acceptance letter]

9 Jan 2025

PONE-D-23-42156R1 

PLOS ONE

Dear Dr. Abajobir, 

I'm pleased to inform you that your manuscript has been deemed suitable for publication in PLOS ONE. Congratulations! Your manuscript is now being handed over to our production team.

Kind regards, 

on behalf of

Professor Ammal Mokhtar Metwally 

Academic Editor

PLOS ONE